# BUILDING SOCIAL WORLD MODEL
# WITH LARGE LANGUAGE MODELS

## ABSTRACT

Understanding and predicting how social beliefs evolve in response to events, ranging from policy changes to scientific breakthroughs, remains a fundamental challenge in social science research. Given that Large Language Models (LLMs) have demonstrated commonsense knowledge and social intelligence, a natural question arises: *Can LLMs be used to model the dynamics of social beliefs following social events?* Addressing this problem can deepen our understanding of community dynamics and inform better decision-making in the real world. In this work, we introduce the concept of the Social World Model (SWM), a general framework designed to capture how social beliefs evolve in response to major events. SWM learns state-transition functions for social beliefs by mining temporal patterns in social data and optimizing evidence lower bound, without the need for explicit human annotations that link events to belief shifts or expensive census data. To evaluate SWM's effectiveness in predicting social belief transitions, we introduce a benchmark, SWM-Bench, derived from real-world Polymarket data. SWM-Bench includes over 300,000 data samples for social belief prediction tasks spanning diverse domains such as politics, sports, cryptocurrency, and elections. Our experimental results show that SWM significantly outperforms traditional time-series models, achieving a 21.56% reduction in RMSE while offering interpretable insights into the underlying mechanisms of social belief dynamics.

## 1 INTRODUCTION

Diverse social beliefs shape different human communities and the future of mankind (Greif, 1994; Bar-Tal, 2000; Zou et al., 2009). Examples of impactful social beliefs include whether AGI will emerge within the next five years (Feng et al., 2024) or who will be the next U.S. president (Barberá González & Lohan, 2024). While some widely accepted beliefs are unlikely to drastically change, other social beliefs are more volatile, shifting dramatically in response to societal events (Campbell, 1986). For example, as is shown in Fig 1, the U.S. presidential election results influence public expectations about the Federal Reserve's policy in December 2024 (Orphanides, 2024), the price of Solana (a cryptocurrency) (Song et al., 2024), and other political events (Tourangbam, 2024). Understanding how social beliefs evolve is crucial for a wide range of societal applications, from forecasting social events (Spilerman, 1975) to improving decision-making in business and economics (Ariely, 1998). This naturally leads to the question: *Can LLMs be used to model the dynamics of social beliefs in response to significant events?*

Modeling the dynamics of social beliefs is a complex challenge for several reasons. First, social beliefs are not independent with each other; they are deeply interconnected. A shift in one belief can ripple across multiple domains, influencing related beliefs simultaneously. This interdependence makes modeling social dynamics particularly intricate. Second, identifying the temporal relationship between a social event and its impact on social beliefs is challenging. The influence of an event may not be immediate or direct, and multiple factors often interact in complex ways to shape public opinion. It requires advanced social reasoning abilities to understand temporal relationships between events and beliefs. Finally, evaluating models of social belief dynamics is challenging due to the lack of well-established tasks or metrics to measure their effectiveness. These challenges underscore the complexity of accurately capturing how social beliefs evolve over time.

Building on these challenges, we introduce the *Social World Model* (SWM), a general framework designed to predict the social belief dynamics based on different social events. SWM adopts a

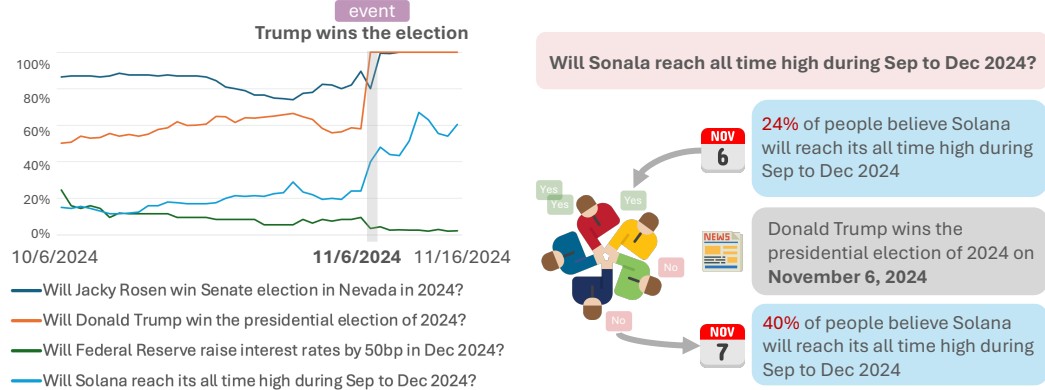

Figure 1: **Social events shape future social beliefs**. Each line tracks social beliefs on a social event over time, collected from Polymarket. A notable social event will create a sudden change in social beliefs. The Social World Model aims to predict how these social beliefs will evolve based on historical social beliefs and a (hypothetical) social event.

state-transition style world model framework, modeling the probability $P(\mathbf{S}_t \mid \mathbf{S}_{<t}, e_t)$, where *state* $\mathbf{S}_{<t}$ represents the community's historical social beliefs, $\mathbf{S}_t$ represent the future social belief at time $t$, and *action* $e_t$ denotes a (hypothetical) social event that impact the future social belief at specific time $t$. A concrete use case of a social world model is shown in Fig 1; suppose today is 11/05/2024, based on the historical social beliefs over different events $\mathbf{S}_{<t}$, stored as a multi-variate time series, conditioned on an action $e_t$, Trump wins the election on 11/06/2024, which is a hypothetical social event, can we predict the remaining future beliefs $\mathbf{S}_t$ on 11/06/2024?

We argue that advanced social reasoning (Sap et al., 2020) and implicit social knowledge are crucial in shaping this transition function and that LLMs are essential for this task, as their large-scale training on human data enables them to capture these underlying social dynamics (Halawi et al., 2024). To leverage this capability, we develop a training framework based on LLMs that predicts potential belief shifts and infer the likely events driving these changes. Our approach utilizes purely time-series social belief data and incorporates posterior guidance to enhance social event inference.

Beyond developing SWM, we introduce SWM-Bench, the first benchmark for real-world social belief prediction that is tailored for both LLMs and time-series models. We collected extensive real-world time-series event data from sources such as daily win rates on Polymarket, one of the world's largest social prediction platforms. In SWM-Bench, each topic on Polymarket represents a social belief, and every belief change over a given threshold is treated as a natural event. SWM-Bench includes social belief prediction tasks spanning diverse domains, including elections, politics, cryptocurrency, and sports. This comprehensive benchmark enables a thorough evaluation of SWM 's ability to model and predict belief shifts across different domains.

Based on the experiments, we find that utilizing the proposed SWM to conduct social belief predictions provides better results on SWM-Bench compared with time-series-based methods or LLM-based methods. Specifically, the proposed social world model outperforms baselines by 10∼39% in our real-world experiments. Overall, our contributions include: (1) We are one of the first to formally propose the concept of the social world model, defining the task of predicting shifts in social belief given specific social events; (2) We collected a novel benchmark from PolyMarket named SWM-bench to quantify the performance; (3) We introduce a belief-change modeling method that marginalizes over diverse social events while decoupling the task into two components: social reasoning and social world modeling; (4) We propose a novel training framework using a posterior-guided social reasoner and an ELBO-based optimization objective for both components.

## 2 RELATED WORKS

**World model**. A world model can be formulated as state transition probabilities, which characterize a generative mechanism of how the world state changes after an agent's actions (Hu & Shu, 2023). The most recent solution considers each state in the accurate dynamics model in reaction to control in the pixel space (*e.g.* video frame) instead of the low-dimensional space. Each action is also described in the text as human instructions. A world model can be a task-specific simulator for robotics

manipulation or self-driving (Wang et al., 2023). Our proposed social world model simulates societal dynamics. Notably, the social world model represents human beliefs as the state and physical events as actions, forming a dual to the existing world models, where states come from the physical world and actions from human instructions.

**Commonsense social reasoning**. Commonsense social reasoning (Sap et al., 2020) refers to the ability to understand the motivation, results, and emotional reactions (Rashkin et al., 2018) of specific social events. Such understanding relies heavily on the commonsense knowledge of social rules. Understanding social reasoning can conduct better social interaction in our daily life (Sap et al., 2019). Neural symbolic structures or language models (LLMs) are utilized as a knowledge base to conduct social reasoning Zhong et al. (2019). However, commonsense social reasoning mainly focuses on individual-level social events like daily routes or conversation (Tenney et al., 2019) while the social world model focuses more on community-level social events (Petroni et al., 2019) that brings benefits to multiple individuals in the community and change the future of some overall trend at the community level. Complicated consensus and controversy within the community requires more complicated analysis on the community level.

**Social event forcasting**. Event prediction involves forecasting future events based on historical data (Hendrycks et al., 2021; Jin et al., 2020). Traditional approaches rely on semantic information and time-series modeling (Zou et al., 2022). Moreover, language models have been proven to be useful in providing high-quality forecasting results since it is able to conduct social reasoning and identify temporal relationships between events (Halawi et al., 2024; Abolghasemi et al., 2024; Schoenegger & Park, 2023). However, the social world model has a different target compared with event forecasting. Instead of targeting forecasting the final results of the events (Woo et al., 2024), the social world model tries to model the community and predict the change of public opinions after the occurrence of certain events. The final results of the events are highly uncertain and hard to predict even with crowdsourcing power. Moreover, some outbreaking news like natural disasters and political changes are considered as unpredictable. However, people's reactions towards social events can be valuable and predictable with a high-quality modeling of the community itself.

## 3 PRELIMINARIES

**Data selection for social modeling**. Our society is a highly dynamic and complex system where individuals with diverse cultural, political, and educational backgrounds often hold different opinions about the same social events. Admittedly, selecting the ideal data for modeling our society is a critical challenge, as available social opinion data are often noisy, vague, and inauthentic. Our insight is that free markets, where rational individuals freely exchange transactions to maximize self-utilities, serve as a classical example of aggregated public opinions; therefore, we argue that market-related data imply a high-quality social opinions data source with clearer signals and more authentic opinions compared to other popular dataset options. Among the publicly available market data, Polymarket, one of the largest online prediction markets, offers a comprehensive platform that effectively captures a wide range of social beliefs and is being faithfully traded among the participants.

**Polymarket prediction market**. Polymarket markets span multiple domains including politics, technology, cryptocurrency, sports, and more, making it a versatile source for studying social beliefs. Several key features make Polymarket particularly suited for analyzing the dynamics of social belief changes: (1)*Uncertainty and popularity*: Markets listed on Polymarket inherently involve uncertain outcomes. Events with widely predictable outcomes or generally agreed-upon knowledge typically do not appear on Polymarket because they lack potential financial incentives (Wolfers & Zitzewitz, 2004). (2)*Diversity and scale*: Participants in Polymarket come from diverse backgrounds, bringing varied perspectives into their predictions. This broad participant base ensures that shifts in predictions effectively reflect changes in general societal consensus (Arrow et al., 2008). (3)*Investment-driven quality*: The financial stakes involved ensure participants' decisions are deliberate and thoughtful rather than arbitrary or influenced by random noise (Manski, 2006). The monetary investment component thereby improves the reliability and quality of the collected social belief data. Consequently, Polymarket serves as an exemplary platform for capturing and analyzing the dynamic shifts in social beliefs (Ottaviani & Sørensen, 2007), making it particularly effective for research on opinion formation, public decision-making processes, and societal forecasting.

**Social belief**. We consider a *social belief* as a collective opinion on a yes/no question referring to a future time $T$. Concretely, each market in the Polymarket data can be considered as an observation

point about social belief. Formally, at time $t$ (with $t < T$), each social belief is represented by an ordered pair $s_t = (q_T^i, a_t^i)$, where $q_T^i$ is a yes/no question about an event or outcome by the future time $T$, and $a_t^i$ is the community's numerical expectation (*e.g.*, the fraction of "Yes" votes). For instance, the question "Will OpenAI release GPT-5 in February 2025?" can be represented by $q_T^i$, and the community's aggregated belief $a_t^i$ is the average of the individual votes at time $t$. In Polymarket, real-time prices (win rates) often serve as a proxy for these collective beliefs.

**Social event**. A *social event* is a real-world occurrence at a specific time $t$. Concretely, a social event is referred to a real-world news as specific time. We denote such an event by $e_t = (x, t)$, where $x$ is a textual description of the occurrence, and $t$ is a timestamp. For example, *"Donald Trump is inaugurated on January 20, 2025, as the 47th President of the United States"* could be annotated as $e_t$. Some social events are surprising or especially impactful, causing major shifts in social beliefs, while others are routine or expected and thus have minimal effect.

**Belief space**. A *belief space* at time $t$ is the collection of *all* social beliefs held by the community at that time, which can be estimated by all data from PolyMarket. Formally, a social state is a set of social beliefs $\mathbf{S}_t = \{s_t^1, s_t^2, \ldots, s_t^n\}$. A larger set of social beliefs ($n$ being large) typically yields a more detailed depiction of the community's collective stance. As new information or events emerge, a community may update its beliefs, causing $\mathbf{S}_t$ to evolve over time.

**Event space**. The *event space* at time $t$ is the set of all social events that occur at that time. Formally, we define $\mathbf{E}_t = \{e_t^1, e_t^2, \ldots, e_t^m\}$. As $\mathbf{E}_t$ encompasses all real-world events at time $t$, it can be quite large, with each event carrying a potential influence on the community's beliefs.

# 4 PROPOSED CONCEPT: SOCIAL WORLD MODEL

The *social world model* seeks to predict how social states $\mathbf{S}_t$ evolve over time in response to social events $e_t$ (both real-world and hypothetical). This model captures the dynamic temporal relationships between these events and social states, aiming to understand and simulate the complex behaviors of human social communities. For instance, when a significant social event like policy announcement $e_t$ occurs at time $t$, the current state of social state $\mathbf{S}_t$ reflects varying levels of support, concern, or other sentiments related to the event's implications for future outcomes. Concretely, we parameterize the modeling approach by $P_\theta(\mathbf{S}_t \mid \mathbf{S}_{<t}, e_t)$, where $\theta$ represents the model parameters that capture the relationships and dynamics. The concept of social world model can be formally defined as below:

**Definition 1** (*Social World Model* (SWM)). *Let $\mathbf{S}_t = \{s_t^1, s_t^2, \ldots, s_t^n\}$ be the set of social beliefs at time $t$. Let $\mathbf{S}_{<t} = \{\mathbf{S}_{t-1}, \mathbf{S}_{t-2}, \ldots, \mathbf{S}_{t-k}\}$ be the collection of historical social states. Finally, let $e_t$ be any hypothetical social event that occurs at time $t$. The social world model is defined as the following state-transition function: $\mathbf{S}_t \sim P_\theta(\mathbf{S}_t \mid \mathbf{S}_{<t}, e_t)$, where $\theta$ are the model parameters.*

**Comparison to existing world models definition**. In classical world models (Hu & Shu, 2023), one often sees an equation of the form $\mathbf{S}_{t+1} \sim P_\theta(\mathbf{S}_{t+1} \mid \mathbf{S}_t, a_t)$, where $\mathbf{S}_t$ is the current world state (*e.g.*, a video frame), $a_t$ is an action taken by an agent (*e.g.*, an human instruction), and $\mathbf{S}_{t+1}$ is the next world state after the action. This assumes a Markovian property (Norris, 1998), whereby only $\mathbf{S}_t$ (rather than all previous states) is needed. By contrast, the social world model uses $\mathbf{S}_{<t}$ (a history of states) and a social event $e_t$ that can be described in text form. Historical information can play a key role in social belief prediction, making the evolution of social states $\mathbf{S}_t$ beyond Markovian property. Additionally, the social world model instantiates actions as hypothetical social events, so that it can be trained with historical data without external supervision.

# 5 BUILDING SOCIAL WORLD MODEL WITH LLMS

Parameterizing the social world model $P_\theta(\mathbf{S}_t \mid \mathbf{S}_{<t}, e_t)$ with LLMs faces two main challenges: (1) collecting paired state-action training data $\{(\mathbf{S}_{<t}^{(i)}, \mathbf{S}_t^{(i)}, e_t^{(i)})\}_{i=1}^N$ is hard and (2) the large size of event space $\mathbf{E}_t$. To address these challenges, we observe that events can be categorized based on their impact on social beliefs: (1) *consensus events* are those with widely expected outcomes, such as routine daily occurrences like sunrise; (2) *controversial events* are those with uncertain or contested outcomes, like presidential elections. These events often significantly shift social beliefs as they either confirm or challenge existing expectations within communities. Overall, in Section §5.1 and §5.2, we propose two complementary strategies to account for belief changes for social world modeling. We design an inference algorithm based on its hybrid version in Section §5.5.

Figure 2: **Posterior-guided training and social belief prediction for both social reasoner and social world model**. $P_\eta$ stands for social reasoner, $Q_\phi$ stands for posterior-guided social reasoner, $P_\theta$ stands for social world model. $P_\theta$ and $P_\eta$ are jointly used for social belief prediction. Each piece of news collected from the real world at the date of Nov 6, 2024, is considered a social event $e_t$. A social world model $P_\theta$ can be independently utilized given hypothetical events and social states, but for social belief prediction, it requires the help from $P_\eta$. $P_\theta$, $P_\eta$, and $Q_\phi$ are parameterized with different LLMs separately.

## 5.1 Designing SWM without Event Modeling

Within the event space $\mathbf{E}_t$, consensus events make up the majority. Humans, as intelligent agents, actively observe the world and therefore have already incorporated these consensuses in the prior social states $\mathbf{S}_{<t}$. Therefore, we assume consensus social events can be ignored in the condition and the conditional probability can be simplified as: $P(\mathbf{S}_t \mid \mathbf{S}_{<t}, e_t) \approx P(\mathbf{S}_t \mid \mathbf{S}_{<t})$. The setting over consensus events allows us to train a social world model using only social state data $\{(\mathbf{S}_{<t}^{(i)}, \mathbf{S}_t^{(i)})\}_{i=1}^N$, eliminating the need for explicit event annotations.

Furthermore, since each social belief $s_t \in \mathbf{S}_t$ is determined simultaneously at time $t$, we assume that each belief is conditionally independent given the historical social states. Reasons why we think such an assumption is necessary in this case is in Appendix §G. This allows the joint probability to be factorized, giving us the first learning objective of SWM in this case:

$$P(\mathbf{S}_t \mid \mathbf{S}_{<t}) \approx \prod_{s_t \in \mathbf{S}_t} P_\theta(s_t \mid \mathbf{S}_{<t}). \tag{1}$$

## 5.2 Designing SWM with Event Modeling

More rigorously, our social world model should account for both consensus and controversial events, where controversial events could dramatically alter social states $\mathbf{S}_t$. Here, we consider the event space as $\mathbf{E}_t = \{e_t^\emptyset, e_t^1, \cdots, e_t^m\}$, where $e_t^\emptyset$ compress all routine, predictable events into one event. For all other controversial events $\{e_t^1, \cdots, e_t^m\}$ that can cause surprising changes, we need to explicitly model their impact through $P(\mathbf{S}_{t+1} \mid \mathbf{S}_t, e_t)$ in a SWM.

**Event as latent variable**. Due to the lack of paired state-action training data $\{(\mathbf{S}_{<t}^{(i)}, \mathbf{S}_t^{(i)}, e_t^{(i)})\}_{i=1}^N$, we treat each social event $e_t$ as latent variables that drive temporal changes. This allows us to marginalize the social world model to $P(\mathbf{S}_t \mid \mathbf{S}_{<t})$ with event distribution:

$$P(\mathbf{S}_t \mid \mathbf{S}_{<t}) = \sum_{e_t \in \mathbf{E}_t} P_\theta(\mathbf{S}_t \mid \mathbf{S}_{<t}, e_t) P_\eta(e_t \mid \mathbf{S}_{<t}) \tag{2}$$

This equation consists of two components: a social world model $P_\theta$ that predicts event impacts on social states and a social reasoner $P_\eta$ that evaluates event importance. For example, when considering presidential election results as a social event $e_t$, the social world model predicts its impact on cryptocurrency prices while the social reasoner assesses the election's relevance to the crypto market.

**Posterior guidance**. Due to the vast diversity of events, $|\mathbf{E}_t|$ can be extremely large, making the direct calculation of Equation 2 impractical for training. Thus, we observe that $P_\eta$ is sharply distributed since each social belief would not be related to a large number of events, consequently, we can switch our optimization target with the Evidence Lower Bound (ELBO) (Kingma et al., 2013) for each $P(s_t^i \mid \mathbf{S}_{<t})$:

$$\log P(s_t \mid \mathbf{S}_{<t}) \geq \mathbb{E}_{e_t \sim Q_\phi(\cdot \mid \mathbf{S}_{<t}, s_t)} \Big[ \log P_\theta(s_t \mid \mathbf{S}_{<t}, e_t) \Big] - D_{\mathrm{KL}} \Big[ Q_\phi(e_t \mid \mathbf{S}_{<t}, s_t) \,\|\, P_\eta(e_t \mid \mathbf{S}_{<t}) \Big]. \tag{3}$$

where $Q_\phi$ is a posterior-guided social reasoner that identifies related events driving changes in social states, and similar to Equation 1 we assume each belief is conditionally independent. For example, $Q_\phi$ is responsible for identifying which events (like a highly related political announcement) triggered a sudden shift in policy expectations.

## 5.3 COMPONENTS OF SWM FOR TRAINING

In the following parts, we discuss the implementation of three components: (1) posterior-guided social reasoner $Q_\phi$; (2) social reasoner $P_\eta$; (3) social world model $P_\theta$.

**Posterior-guided social reasoner** $Q_\phi$. The posterior-guided social reasoner $Q_\phi$ operates on a limited event space $\mathbf{E}_t$ and leverages future data during training to provide sharp, accurate estimates of important events. For example, given known presidential election results, $Q_\phi$ is responsible for reasoning and identifying which social events were most influential and critical in shaping the outcome. For simplicity, here we approximate $\mathbf{E}_t$ to be the top-$k$ most popular news that occurs every day. This requires multi-step temporal reasoning with social knowledge to understand temporal relationships between events and belief changes. Due to the lack of training data for $Q_\phi$, we leverage the in-context learning and parametric knowledge of state-of-the-art LLMs (specifically gpt-4o[1]) to score event importance.

**Social reasoner** $P_\eta$. The key distinction between $Q_\phi$ and $P_\eta$ is their access to information: $Q_\phi$ can use future social beliefs $\mathbf{S}_t$ during training, while $P_\eta$ cannot. Such lack of information makes the social reasoning tasks much harder. $P_\eta$ is responsible for predicting which events are likely to impact future social states, requiring strong social reasoning capabilities as well. We optimize the social reasoner model $P_\eta$ by deriving and minimizing the reverse KL divergence with the posterior-guided reasoner.

$$D_{\mathrm{KL}}\Big[Q_\phi\big(e_t \mid \mathbf{S}_{<t}, s_t\big) \,\|\, P_\eta(e_t \mid \mathbf{S}_{<t})\Big] = \sum_{e_t \sim Q_\phi} Q_\phi(e_t \mid \mathbf{S}_{<t}, s_t)(\log Q_\phi(e_t \mid s_t, \mathbf{S}_{<t}) - P_\eta(e_t \mid \mathbf{S}_{<t}))$$

(4)

where the importance scores from $P_\eta$ are again weighted by $Q_\phi$, following a probabilistic implication: if $Q_\phi$ assigns high importance to an event, $P_\eta$ should do likewise. Intuitively $Q_\eta$ teaches advanced social knowledge that can be useful for measuring the importance of one event based on its experience.

**Social world model** $P_\theta$. Our target model $P_\theta$ is optimized with guidance from $Q_\phi$, using the posterior-weighted expectation:

$$\mathbb{E}_{e_t \sim Q_\phi(\cdot \mid \mathbf{S}_{<t}, s_t)}\Big[\log P_\theta(s_t \mid \mathbf{S}_{<t}, e_t)\Big] = \sum_{e_t \in \mathbf{E}_t} Q_\phi(e_t \mid \mathbf{S}_{<t}, s_t) \log P_\theta(s_t \mid \mathbf{S}_{<t}, e_t)$$

(5)

The social world model weights each event $e_t$ by $Q_\phi$ during training, enabling it to focus on learning true temporal relationships between events and state changes. This training approach, guided by $Q_\phi$'s distribution, also enhances the model's performance when using $P_\eta$ for social belief prediction.

**Social belief retriever**. After addressing the large size of $\mathbf{E}_t$, the remaining challenges for using LLMs to parameterize these three components are LLM's limited context window and the large size of $|\mathbf{S}_t|$. Therefore, a social belief retriever $\mathrm{Retro}(q, \mathcal{D})$ that finds $d \in \mathcal{D}$ related to query $q$ is required to retrieve useful and related historical social beliefs to augment the modeling process. For example, when implementing the social world model $P_\theta$, we use:

$$P_\theta(\mathbf{S}_t \mid \mathbf{S}_{<t}, e_t) \approx \prod_{i=1}^{n} P_\theta(s_t \mid \mathrm{Retro}(s_{t-1}, \mathbf{S}_{<t}), e_t)$$

(6)

Similar use of a retriever is often required for the training and inference of social reasoner $P_\eta$ and $Q_\phi$ as well to overcome the large size of $|\mathbf{S}_t|$, *e.g.*, Equation 5 becomes

$$\sum_{e_t \in \mathbf{E}_t} Q_\phi(e_t \mid \mathrm{Retro}(s_{t-1}, \mathbf{S}_{<t}), s_t) \log P_\theta(s_t \mid \mathrm{Retro}(s_{t-1}, \mathbf{S}_{<t}), e_t)$$

(7)

## 5.4 SWM TRAINING ALGORITHM

Based on the previous discussion of model components for training, the detailed training algorithm for our social world model can be described as an algorithm below:

---

[1]We specifically use `gpt-4o-2024-08-06` API

## 5.5 SWM INFERENCE ALGORITHM

**Algorithm: SWM Training Pass**

**Input:** Historical states $\mathbf{S}_{<t}$, belief $s_t$, possible events $\mathbf{E}_t$, generative model $f_\theta$, prior $P_\eta$, posterior $Q_\phi$
**Output:** Loss $\mathcal{L}$
$\mathbf{S}'_{<t} \leftarrow \text{Retro}(s_{t-1}, \mathbf{S}_{<t})$
$q_t(\cdot) \leftarrow Q_\phi(\cdot \mid \mathbf{S}'_{<t}, s_t)$
$p_t(\cdot) \leftarrow P_\eta(\cdot \mid \mathbf{S}'_{<t})$
$\mathcal{L}_{\text{KL}} \leftarrow D_{\text{KL}}(q_t \| p_t)$
$\hat{s}_t \leftarrow \sum_{e_t \in \mathbf{E}_t} q_t(e_t) \cdot f_\theta(\mathbf{S}'_{<t}, e_t)$
$\mathcal{L} \leftarrow \text{MSE}(\hat{s}_t, s_t) + \mathcal{L}_{\text{KL}}$
**return** $\mathcal{L}$

After training with time-series data pairs for social belief prediction, we obtain a social world model $P_\theta(\mathbf{S}_t \mid \mathbf{S}_{<t}, e_t)$ and a social reasoner $P_\eta(e_t \mid \mathbf{S}_{<t})$. During inference, we jointly use both models with and without explicit event modeling to adapt to a generic scenario. If a provided event $\hat{e}_t$ is considered to be not related to the current social state and has importance score lower than threshold $\delta$, we classify it as $e_t^\emptyset$ and the model degenerates to $P_\theta(\mathbf{S}_t \mid \mathbf{S}_{<t})$. The joint inference process is written as:

$$\mathbf{S}_t \sim \begin{cases} P_\theta(\mathbf{S}_t \mid \mathbf{S}_{<t}) & \text{if } P_\eta(e_t \mid \mathbf{S}_{<t}) < \delta, \\ P_\theta(\mathbf{S}_t \mid \mathbf{S}_{<t}, e_t) & \text{otherwise.} \end{cases}$$

More discussion related to the analysis of inference algorithm (including $\delta$) is in Appendix §H.

## 6 EXPERIMENTAL SETTINGS

**Data settings**. We collect all market data from Polymarket from the beginning to January 05, 2025, as shown in Table 4. The dataset is manually split into five domains based on provided Polymarket tags: Politics, Sports, Crypto, Election, and Other. For testing, we focus on time points $t$ where social beliefs show dramatic changes ($|s_t - s_{t-1}| > 0.25$), resulting in 360 samples. These sudden shifts indicate significant external event influences, presenting a more challenging scenario than standard time-series prediction. We denote this subset as SWM-Bench (test-hard), distinct from the complete SWM-Bench (test).

**Task settings**. We evaluate our social world model on social belief prediction tasks. Each task requires predicting social belief $s_t$ given historical social states $\mathbf{S}_{<t}$ containing $n$ beliefs and an event space $\mathbf{E}_t$ with $k$ social events, where $s_{<t}$ is included in the historical states. Such tasks require joint usage of social reasoner and social world model.

**Baseline settings**. We compare SWM against two families of methods: (i) time-series forecasting models—LSTM (Hochreiter, 1997), Transformer (Vaswani, 2017), Autoformer (Wu et al., 2021), Informer (Zhou et al., 2021), Reformer (Kitaev et al., 2020), the linear baseline DLinear (Su, 2022), and ChatTime-base/chat (Wang et al., 2025)—each trained within domain on SWM-Bench; and (ii) LLMs with time-series conditioning (LLM w/TS), where closed-source models gpt-4o and o3-mini are prompted with a standardized template that concatenates the recent domain-specific time series and a brief event description, without fine-tuning. All baselines are evaluated under identical protocols, and we report RMSE/MAE per domain in Table 1.

**Metric settings**. We evaluate social belief prediction using two standard regression metrics: Root Mean Square Error (RMSE) and Mean Absolute Error (MAE). RMSE is sensitive to larger errors, while MAE provides the average absolute difference between predicted and actual values.

## 7 EXPERIMENTAL RESULTS

In this section, we present our main results for SWM-Bench in Table 1 and analyze the insights.

**SWM outperforms baselines**. Our proposed SWM significantly outperforms all baselines (including time-series models and LLMs). Time-series baselines perform poorly as they can only rely on historical patterns, making them inadequate for predicting sudden changes. These models do not have access to social knowledge to facilitate their predictions. While LLM baselines better capture temporal patterns, they still fall short compared to SWM. By incorporating social event modeling as latent variables, SWM effectively captures the reason behind potential changes and demonstrates superior performance across all domains.

**Different domains show different trends with SWM.**. Our SWM model delivers significant performance improvements across multiple domains. In politics, it reduces RMSE by 13.72% and MAE by 8.19%. In sports, the gains are smaller, 9.95% in RMSE and 11.31% in MAE, likely due to the high uncertainty of sports-related social events, making daily news an unreliable signal. The

Table 1: **Evaluation results for social belief prediction on SWM-Bench (test-hard).** Each model is trained and tested with data of the same domain. The improvement rate is calculated between the best baseline and SWM. LLM w/TS means including time-series data inside the prompt.

| Model | Politics | | Sports | | Crypto | | Election | | Other | |
|---|---|---|---|---|---|---|---|---|---|---|
| | RMSE | MAE | RMSE | MAE | RMSE | MAE | RMSE | MAE | RMSE | MAE |
| LSTM | 0.413 | 0.396 | 0.368 | 0.360 | 0.431 | 0.403 | 0.440 | 0.424 | 0.377 | 0.362 |
| Transformer | 0.414 | 0.396 | 0.360 | 0.350 | 0.433 | 0.405 | 0.437 | 0.422 | 0.378 | 0.363 |
| Autoformer | 0.398 | 0.321 | 0.382 | 0.310 | 0.383 | 0.308 | 0.418 | 0.340 | 0.396 | 0.327 |
| Informer | 0.397 | 0.319 | 0.381 | 0.310 | 0.384 | 0.310 | 0.415 | 0.336 | 0.398 | 0.329 |
| Reformer | 0.398 | 0.318 | 0.381 | 0.309 | 0.383 | 0.309 | 0.415 | 0.335 | 0.394 | 0.325 |
| DLinear | 0.396 | 0.317 | 0.380 | 0.307 | 0.383 | 0.309 | 0.414 | 0.334 | 0.393 | 0.322 |
| ChatTime-base | 0.392 | 0.336 | 0.305 | 0.271 | 0.393 | 0.345 | 0.436 | 0.386 | 0.398 | 0.356 |
| ChatTime-chat | 0.387 | 0.338 | 0.299 | 0.266 | 0.421 | 0.389 | 0.451 | 0.408 | 0.396 | 0.357 |
| gpt-4o w/TS | 0.428 | 0.403 | 0.369 | 0.356 | 0.421 | 0.391 | 0.438 | 0.418 | 0.389 | 0.372 |
| o3-mini w/TS | 0.415 | 0.388 | 0.387 | 0.368 | 0.419 | 0.388 | 0.444 | 0.423 | 0.378 | 0.368 |
| SWM-0.5B | 0.339 | 0.292 | 0.292 | 0.257 | 0.271 | 0.225 | **0.327** | **0.283** | 0.255 | 0.201 |
| SWM-1.5B | 0.337 | **0.291** | 0.295 | 0.258 | 0.270 | 0.224 | 0.331 | 0.291 | 0.254 | 0.202 |
| SWM-3B | 0.333 | 0.288 | 0.296 | 0.258 | 0.264 | 0.214 | 0.330 | 0.286 | 0.250 | 0.196 |
| SWM-7B | **0.334** | 0.294 | **0.270** | **0.236** | **0.253** | **0.212** | 0.329 | 0.289 | **0.249** | **0.196** |

Table 2: **RMSE on SWM-Bench (test-hard) with different social reasoners**. Marginal uses the social reasoner; ELBO uses the posterior-guided reasoner.

| Method | Poli. | Sports | Crypto | Elec. | Other |
|---|---|---|---|---|---|
| Marginal | 0.266 | 0.241 | 0.256 | 0.288 | 0.340 |
| ELBO | 0.326 | 0.312 | 0.276 | 0.324 | 0.283 |

Table 3: **RMSE on SWM-Bench (test)**. Most data in SWM-Bench (test) change smoothly. We conduct evaluations with SWM on the complete time-series data besides significant change.

| Model | Poli. | Sports | Crypto | Elec. | Other |
|---|---|---|---|---|---|
| Transformer | 0.0193 | 0.0504 | 0.0435 | 0.0274 | 0.0177 |
| SWM | 0.0435 | 0.0661 | 0.0714 | 0.0597 | 0.0628 |

crypto domain shows the most significant improvements, with RMSE and MAE reductions of 33.91% and 30.61%, respectively, this is mainly because crypto-related events are highly sensitive to the occurrence of related news and event modeling works the best. For election forecasting, the model reduces RMSE by 21.06% and MAE by 15.49%, reflecting its strength in analyzing human-centric data. The other category also shows robust enhancements, with reductions of 29.13% in RMSE and 39.21% in MAE, demonstrating SWM 's broad applicability.

Overall, SWM integrates social events into its modeling framework, addressing the limitations commonly found in traditional time-series models for social belief prediction. These results strongly advocate for adopting SWM in diverse and complex social tasks across various fields.

## 8 ABLATION STUDY

**Ablation on model size**. One key factor influencing social belief prediction performance is the capacity of the underlying language model. As shown in Table 1, our social world model is fine-tuned on Qwen2.5-Instruct models of varying sizes: 0.5B, 1.5B, 3B, and 7B. The predictive accuracy consistently improves as the model size increases, suggesting that stronger base models lead to better social belief forecasting. This highlights the critical role of model capacity in enhancing predictive performance for socially grounded tasks. We include more ablation study results on model families in Appendix §D.

**Ablation on window size**. Another key factor in the social world model $P_\theta(\mathbf{S}_t \mid \mathbf{S}_{<t}, e_t)$ is the amount of historical data used to model $\mathbf{S}_{<t} = \{\mathbf{S}_{t-1}, \ldots, \mathbf{S}_{t-k}\}$. A larger $k$ includes more past data for prediction. However, as shown in Fig 3, adding more time-series data does not necessarily improve performance, particularly for predicting dramatic changes. It indicates that for sudden change prediction case, historical data only has limited use.

**Ablation on event space size**. The size of the event space $\mathbf{E}_t$ plays a crucial role in training the social world model. When optimizing the ELBO loss, training is performed under $\sum_{e_t \in \mathbf{E}_t} Q_\phi(e_t \mid \mathbf{S}_{<t}, s_t^i) \log P_\theta(s_t^i \mid \mathbf{S}_{<t}, e_t)$, where $\mathbf{E}_t$ determines the number and diversity of events $d_t$ used for

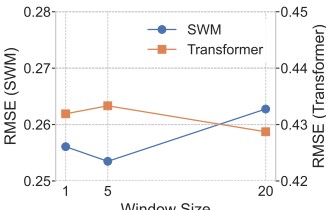
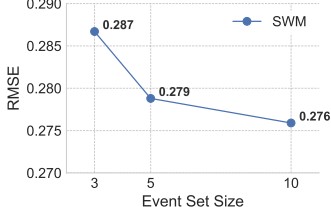
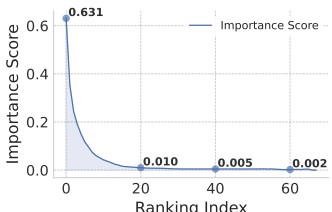

Figure 3: **Ablation study on window size of historical social state $S_{<t}$.** Results are from the Crypto category.

Figure 4: **Ablation study on the size of event space $E_t$.** Results are collected from the Crypto category.

Figure 5: **Event score distribution from $Q_\phi$.** Each point represents an average score for a ranking position.

training. As shown in Fig 4, a larger event space allows the model to incorporate more social events. We observe that reducing the latent variable size from 10 to 5 increases the RMSE for the Crypto category from 0.2759 to 0.2788. Further reducing it to 3 increases RMSE to 0.2867. This suggests that incorporating a greater variety of events during training improves generalization, leading to better performance on unseen events during testing.

## 9 DISCUSSION

**Q1. Is the posterior-guided social reasoner useful?** The posterior-guided social reasoner serves two key roles in ELBO loss training: (1) providing soft labels for KL divergence calculation for the training of the social reasoner, and (2) generating expectation targets to reweight each event during the training of the social world model. As shown in Fig 5, the posterior-guided social reasoner produces a sharp distribution indicating the importance of social events for the current social state. On average, the top-ranked social event in $E_t$ has an importance score of 0.62, while the 10th-ranked event has only 0.04. It allows efficient training for both social world models and social reasoners by selecting top-10 events in the event space. We provide a detailed example in Appendix §J, which illustrates how indirect connection between events is detected by posterior-guided social reasoner and included in the training process. Additionally, as shown in Table 2, the posterior-guided social reasoner provides more accurate predictions across multiple domains. This leads to improved final results when weighting with the posterior distribution rather than the prior social reasoner. These findings indicate that the posterior-guided social reasoner effectively enhances the training process by providing more reliable event weighting, ultimately improving the performance of the SWM.

**Q2. What do social worlds models use for prediction?** As shown in Table 3, when tested on all available sequential data in the test split, our social world model performs worse than pure time-series models such as Transformer, which rely solely on time-series data. However, in cases of sudden changes, the social world model outperforms these baselines. This suggests that while our model may not match the fluency of pure time-series models in continuous trend prediction, it effectively leverages social knowledge from LLMs to improve predictions in dynamic scenarios. To further analyze the role of social events, we conduct an ablation study by removing text-based event information from the social world model during training. This results in an RMSE increase from 0.2759 to 0.2857, indicating that incorporating text-based social event information enhances prediction accuracy. A detailed case study is presented in Appendix §J illustrates how the social world model incorporates relevant events, such as ETF filings and Bitcoin price surges, to adapt its predictions. This demonstrates that the social world model can dynamically adapt to evolving social events, leading to more responsive and accurate forecasts.

## 10 CONCLUSION

In this work, we introduce the concept of the social world model (SWM), which we define as a state-transition function that models social states based on specific social events. Since paired data for states and events is scarce, we treat events as latent variables and optimize the social world model with event modeling using ELBO loss. To evaluate the effectiveness of our approach, we design social belief prediction tasks and construct a benchmark dataset, SWM-Bench, containing over 300,000 datapoints. Experimental results show that SWM significantly outperforms LLM and time-series baselines, particularly in predicting sudden shifts in social belief states. Looking ahead, we believe that social world models have broad real-world applications.

## REPRODUCIBILITY STATEMENT

We include our full codebase in the supplementary material, and we will put the code on GitHub. In addition, we described the overall pipeline of our method in detail in Section §6 and Appendix §B and §C, facilitating readers to reproduce the work.

## ETHICS STATEMENT

Our work models aggregate social beliefs using publicly available, topic-level time series (e.g., Polymarket markets). We do not collect or infer personally identifiable information, and we operate at the level of beliefs about events (elections, crypto, sports) rather than individual users, in accordance with platform terms. The Social World Model (SWM) is designed to forecast belief dynamics—not to establish causality; causal interpretations of outputs would be inappropriate. Because both LLMs and source platforms can encode historical biases or reflect unequal media attention, we emphasize transparency and reproducibility by releasing code, documenting data coverage and preprocessing, and reporting uncertainty alongside predictions.

**Potential social impact.** Positively, SWM can help journalists, researchers, and policymakers understand opinion dynamics, perform "what-if" stress tests around salient events, and compare methods on SWM-Bench, which bridges time-series modeling and LLM-based reasoning. However, risks include misuse for targeted persuasion or market manipulation, amplification of rumor-driven shocks, overconfidence in forecasts in high-stakes settings, entrenchment of upstream biases, and feedback loops whereby public forecasts shift beliefs or prices, degrading calibration and potentially harming less-informed participants. Affected stakeholders include news consumers, retail traders, civil society groups, election observers, and platforms that host belief markets.

**Mitigations and responsible use.** We (i) restrict inputs to aggregate, public data and disallow individual-level predictions; (ii) document limitations, known biases, and distribution-shift tests; (iii) publish calibrated uncertainty (e.g., intervals, reliability diagnostics) and support abstention when confidence is low; (iv) avoid micro-targeted outputs (e.g., rate-limit or aggregate forecasts) and provide delayed or batched reporting to reduce feedback effects; (v) conduct red-teaming with counterfactual events and adversarial perturbations; (vi) include model cards and data statements detailing provenance, coverage, and compute; and (vii) add usage terms prohibiting deployment for electoral manipulation, deceptive practices, or harms to vulnerable populations. Environmental impact is considered by reporting compute and preferring efficient training/evaluation settings.

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

## A  THE USE OF LARGE LANGUAGE MODELS (LLMS)

We used ChatGPT as a writing assistant to help us write part of the paper. Additionally, we utilize the power of CodePilot to help us code faster. However, all the AI-generated writing and coding components are manually checked and modified. There is no full AI-generated content in the paper.

## B  ASSET DETAILS

### B.1  CODE AND DATA OPEN-SOURCE

We will release both our code and the complete dataset upon acceptance of the paper. The dataset will be shared under the Open Database License (ODbL), allowing use, sharing, and adaptation with proper attribution and requiring that derivative datasets remain open. Our data is collected from publicly accessible sources, and we ensure full compliance with the terms of those platforms.

### B.2  DATASET DETAILS

We outline below the technical details of our dataset collection, which includes both news data and social belief data from PolyMarket.

**News data collection**    We use the API service provided by `thenewsapi`, specifically `https://api.thenewsapi.com/v1/news/headlines`, to collect daily news from 2021 to 2025. The API returns more than 10 significant news headlines and descriptions each day. These news items serve as part of the input for our Social World Model.

**News data statistics**    In total, our news data includes 79,431 news items. The overall available dates range across 1463 days. The average news per day is 54.29. The earliest day included is 01/01/2021 and the latest day included is 01/02/2025.

**PolyMarket data collection**    To collect real-world belief data, we use the publicly accessible API from the PolyMarket platform: `https://gamma-api.polymarket.com`. We gather both metadata (e.g., market title, status, tags) and historical time-series data for each market, using various parameter settings. Our dataset includes all available PolyMarket data prior to January 1, 2025.

**PolyMarket categorization**    Each PolyMarket event includes multiple tags in its metadata that indicate both fine-grained and coarse-grained topic classifications. We use these tags to classify each event into one of five broad categories: politics, sports, crypto, election, or other. If a tag corresponding to one of these categories is present, we assign the event to that category for downstream use.

**PolyMarket data splitting**    In PolyMarket data, each event can have multiple options and each option can have one time-series. For example, the event about the NBA championship can include multiple NBA teams and each of them includes a time series about the social belief. Within one event, the time-series data for multiple options are highly correlated with each other and can cause data leakage for testing. Therefore, we directly split all events into train, dev, and test based on 80%, 10%, and 10%. Since we split based on events, we do not guarantee that the market number in train, dev, and test is strictly 8:1:1. Table 4 shows the split number of train/dev/test. Our motivation for conducting market-level splitting is to verify that the SWM has good out-of-distribution capability, potentially being able to generalize to fully unobserved events; therefore, we split the train/test by event. For example, testing whether training on time-series of "Will ETH hit $2000 by EOY? " can generalize to predicting "The Fed raises interest rates by June?". We do not enforce an out-of-time split. Instead, we ensure that no two time-series curves from the same market that are semantically similar appear in both train and test sets with the motivation to test the generalization ability of SWM.

**PolyMarket time-series data**    For more fine-grained details, we provide more details about the statistics about the time-series data from the PolyMarket. Table 5 indicates the average time-series length for each category.

| Category | Train | Dev | Test |
|---|---|---|---|
| Crypto | 895 | 71 | 91 |
| Election | 2433 | 307 | 315 |
| Politics | 4088 | 505 | 487 |
| Sports | 5775 | 625 | 579 |
| Other | 4386 | 553 | 620 |
| **Overall** | 14894 | 1882 | 1884 |

Table 4: **Splitting of the dataset.** We include the market number for each category.

**Hard Subset of PolyMarket Data**  During evaluation, we do not test the model at every time step, as our goal is to capture sudden shifts in social belief. Instead, we select a subset of time steps within each time-series that reflect significant changes ($|s_{t+1} - s_t| \geq 0.25$). The number of selected time steps varies by domain: Crypto includes 26 points, Election 51 points, Other 48 points, Politics 95 points, and Sports 18 points.

| Category | Train | Dev | Test |
|---|---|---|---|
| Crypto | 36.15 | 27.62 | 42.24 |
| Election | 41.28 | 26.57 | 22.33 |
| Other | 30.25 | 37.41 | 26.73 |
| Politics | 34.02 | 49.10 | 47.07 |
| Sports | 17.67 | 10.57 | 9.51 |
| **Overall Bench** | 27.44 | 27.60 | 26.61 |

Table 5: **Time-series length of the dataset.** We include the length of time-series data for each category.

### B.3 DATASET LICENSE

We plan to release our dataset under the Open Database License (ODbL), which permits use, sharing, and modification of the data while requiring proper attribution and that any derivative works remain openly available under the same license. Since our dataset is derived from publicly accessible information on Polymarket, we ensure compliance with Polymarket's Terms of Service by avoiding any restricted activities and including clear attribution in our release materials. This licensing choice supports open research while safeguarding both ethical reuse and legal compliance.

### B.4 MODEL DETAILS

SWM training framework includes (1) a Social Reasoner trained with posterior guidance, (2) a Posterior-Guided Social Reasoner providing pseudo-labels, and (3) a Social World Model trained with pseudo-labels and integrated with the reasoner. All with LLMs. The training pipeline is shown in Figure 2: (1) train reasoner, (2) train predictor, (3) combine for inference.

We select the Qwen model family as our backbone for both social reasoner and predictor mainly due to: (1) Efficiency: Small models are faster and more deployable for real-time trading on PolyMarket. (2) Resource constraints: Our budget allows training only small models. (3) Performance: Qwen models are recognized for strong results with compact sizes. For the posterior-guided social reasoner, we require a state-of-the-art model to generate reliable pseudo-labels. A pilot human evaluation showed that GPT-4o achieves the lowest MSE (0.0234), outperforming DeepSeek-V3 (0.0250), DeepSeek-R1 (0.0830), GPT-o3-mini (0.0234), and Qwen2-72B (0.0309). Thus, we use GPT-4o as the posterior-guided social reasoner for its superior human-aligned performance.

### B.4.1 REASONER MODULES

**Posterior Reasoner**  The posterior reasoner uses a structured prompt passed to the GPT-4o model. The full prompt template is:

---

**Prompt for posterior reasoner**

Analyze market price change causation for date:
Market: {question}
Current Price Change:  {direction} from {current_price:.3f} ({current_date}) to {next_price:.3f} ({next_date}) ({change_pct:.1f}%)
Historical Price Data (Previous 5 days): {historical_data}
News: {news_items}
Task: Rate each news item's likelihood (0-100) of causing this price change.
Format:  Return JSON array of objects with "news_id" and "score" fields.  Example: [{"news_id": 0, "score": 85}, {"news_id": 1, "score": 15}]

---

**Prior Reasoner**  The prior reasoner is a LoRA-adapted causal language model `Qwen2.5-0.5B-Instruct` with a regression head, trained to reproduce the posterior belief distribution obtained above.  The training objective is to match the agent-assigned posterior scores (normalized as a probability distribution $p$) using the model's predicted softmax output $q$ over multiple candidate inputs per group (i.e., per market-day tuple).  The loss function is the KL divergence $\mathrm{KL}(p\|q)$.  The model is trained using a custom Hugging Face `Trainer` class `KLDivergenceTrainer`.  All prior reasoner models are trained using `transformers.TrainingArguments`.

---

**Prompt for prior reasoner**

You are given an event: {market_question}
{market.description}
On {date1}, price(Yes) = {win_rate1}
On {date2}, price(Yes) = {win_rate2}
On {date3}, price(Yes) = {win_rate3}
On {date4}, price(Yes) = {win_rate4}
On {date5}, price(Yes) = {win_rate5}
We want to predict the possibility on {date} based on this news:
Description: {news_description}

Rate how relevant this news is (0–100) to the next day's price.

---

### B.4.2 PREDICTOR MODULES

The predictor model is trained to estimate the aggregated reward for a set of evidence sentences (e.g., filtered news) with respect to their alignment with posterior scores.  For each market-day example, the model receives multiple evidence items with learned weights, and is trained to output a single score that matches the known reward (ground-truth label). The predictor is built on top of a pretrained causal language model (`Qwen2.5-0.5B-Instruct`, `Qwen2.5-1.5B-Instruct`, `Qwen2.5-3B-Instruct`, `Qwen2.5-7B-Instruct`), extended with a lightweight regression head. The predictor is trained with a custom `Trainer` class (`WeightedTrainer`) that handles group-based weighted loss computation.

We argue that prompt design can be a minor part of both our social world model and social reasoner because we conduct training on it. Training makes the model less reliant on the prompt engineering.

---

**Prompt for predictor**

The news description: {news_description}

Please predict the possibility to happen on date {target_date} based on the following news: {news_content}

At date {date1}, its possibility to be 'Yes' to the event is {number1}.
At date {date2}, its possibility to be 'Yes' to the event is {number2}.
At date {date3}, its possibility to be 'Yes' to the event is {number3}.
At date {date4}, its possibility to be 'Yes' to the event is {number4}.
At date {date5}, its possibility to be 'Yes' to the event is {number5}.

Return a single number in [0, 1] for the predicted probability on {target_date}, and a one–two sentence rationale.

---

### B.5 MODEL LICENSE

We include all the licenses for models that we use during training, inference, and data collection:

**Qwen2-0.5B-Instruct** License: Apache 2.0
**Qwen2-1.5B-Instruct** License: Apache 2.0
**Qwen2-3B-Instruct** License: Apache 2.0
**Qwen2-7B-Instruct** License: Apache 2.0
**GPT-4o** License: Proprietary (OpenAI)
We used OpenAI's GPT-4o (gpt-4o-2024-08-06), a proprietary large language model accessible via API (https://openai.com/gpt-4o). Usage complies with OpenAI's Terms of Use (https://openai.com/policies/terms-of-use).

## C EXPERIMENTAL DETAILS

### C.1 COMPUTING SOURCE

For all training experiments, they are conducted on $\leq 4$ A100 80GB GPUs. For its inference experiments, they are conducted on 1 A100 80GB GPUs.

### C.2 TRAINING DETAILS

In our SWM, there are two important components for training: SWM reasoner and SWM predictor.

**SWM reasoner** We fine-tuned the Qwen2.5-0.5B-Instruct [1] checkpoints using batch size 8, with a maximum sequence length 1024 tokens. LoRA config is lora alpha is 32, lora dropout is 0.1, r is 16.

**SWM predictor** We fine-tuned the Qwen2.5-0.5B-Instruct, Qwen2.5-1.5B-Instruct, Qwen2.5-3B-Instruct, Qwen2.5-7B-Instruct checkpoints using batch size 4, with a maximum sequence length 512 tokens. LoRA config is lora alpha is 32, lora dropout is 0.1, r is 16.

### C.3 INFERENCE DETAILS

For model inference, we integrate both the reasoner and the predictor to form a complete Social World Model. Given a time-series input, the reasoner first ranks the news items based on their relevance. These relevance scores are then used as weights to combine the ranked news with the original time-series data. The resulting weighted representation is passed to the predictor, which generates the final prediction.

---

[1]https://huggingface.co/Qwen/Qwen2.5-0.5B-Instruct

## C.4 SIGNIFICANCE TEST

To assess the statistical significance of our results, we conducted multiple training runs of the `Qwen2.5-0.5B-Instruct` reasoner under identical settings. We trained the model independently for 8 runs with different random seeds. A significance test (e.g., paired t-test) between the best-performing configuration and the baselines yielded **p < 0.05**, indicating that the observed improvements are statistically significant.

## D ADDITIONAL RESULTS ON OTHER OPEN-SOURCE LLM FAMILY

Beyond the Qwen family, we also add experiments with the Gemma family, specifically using Gemma-2-2b-it as the backbone for the social world model. This broadens the evaluation of our framework across different open-source LLMs. Based on the Table, we find that even though we utilize the same training receipt, Gemma-2-2b-it does not match the performance of qwen2.5-0.5B-Instruct with the same amount of training.

## E POLYMARKET AS DATA SOURCE

Based on the publicly avaiable informtion, Polymarket's audience is predominantly male, accounting for approximately 73% of users, with 27% female, and the largest segment falling in the 25–34 age range. While detailed data on education, income, or ethnicity is not publicly available, active users are generally tech-savvy, digitally native individuals with strong interests in DeFi, cryptocurrency, predictive analytics, and speculation markets. Geographically, about 32% of visits come from the United States, followed by Germany, Canada, South Korea, and the United Kingdom. Device usage is nearly evenly split, with roughly 53% accessing via desktop and 47% via mobile devices. Such biased user group can potentially make our social world model biased.

## F GENERALIZATION ABILITY OF SWM

We believe our SWM can be generally applied to the analysis of multiple types of social events. The reason is that LLMs have a large amount of world and social knowledge. Even though different social beliefs and social events have different dependencies and hierarchical structures (like predicting basketball game results and predicting the world economy), they can all be handled by an LLM-based model thanks to its strong in-context learning abilities. Therefore, we believe that an LLM-based social world model can be applied to different event-belief relationship analyses.

For our benchmark limited to PolyMarket, although we currently only include its data, given its scale and data richness, we plan to extend our work to other platforms such as Kalshi and Manifold. Each market has different APIs and user bases, making real-time data collection challenging within the rebuttal period. Nonetheless, because these markets also track high-stakes social beliefs, we expect that their reactions to shocking events will mirror those observed in PolyMarket, allowing our approach to generalize effectively.

## G CONDITIONAL INDEPENDENT BELIEF ASSUMPTION

To better train and inference implementation, we adopt a conditional independence assumption. At each time step, the belief space $S_t$ may involve more than 1,000 time series, which far exceeds the context window of current LLMs. Modeling the full joint distribution $P(S_t \mid S_{<t})$ is therefore infeasible. For efficiency, we approximate independence and restrict the conditioning set to the most relevant prior signals: $\text{top}k(S_{<t}) \subseteq S_{<t}$. This approach allows parallel training while keeping the optimization process within the feasible context length of LLMs.

We detect time points where a belief trajectory shows significant shifts using $z$-score thresholding on price changes. For each event, we extract these change points and compute a synchrony score: the fraction of change points in belief $A$ that occur within a short time window $\delta$ of any change point in belief $B$. This reflects the idea that beliefs with simultaneous fluctuations are likely correlated:

Table 6: **Ablation study on model family**. We include performance on Politics and Election tasks with different modal backbone.

| Model | Politics | | Sports | | Crypto | | Election | | Other | |
|---|---|---|---|---|---|---|---|---|---|---|
| | RMSE | MAE | RMSE | MAE | RMSE | MAE | RMSE | MAE | RMSE | MAE |
| Qwen2.5-0.5B-Instruct | 0.3955 | 0.3215 | 0.3351 | 0.2822 | 0.3129 | 0.2564 | 0.4277 | 0.3534 | 0.2728 | 0.2349 |
| gemma-2-2b-it | 0.3834 | 0.2936 | 0.4607 | 0.3982 | 0.5053 | 0.4219 | 0.4256 | 0.3288 | 0.4769 | 0.4203 |

$$s(A, B) = \frac{1}{|T_A|} \sum_{t \in T_A} \mathbf{1}\big[\exists\, t' \in T_B;\, |t - t'| < \delta\big]. \tag{8}$$

Applying this metric to SWM-bench shows that only a small fraction of belief pairs exhibit high change-point synchrony ($> 0.7$), indicating that few beliefs share similar transition dynamics around social events.

## H    INFERENCE ALGORITHM ANALYSIS

Thresholding $\delta$ plays a central role in our hybrid inference design. The event-aware SWM specializes in dramatic changes, while the time-series-focused model handles smooth trends. Incorrect switching between them could misclassify belief changes. To address this, thresholds are dynamically guided by the LLM-based social reasoner: we use higher thresholds for domains like elections and macroeconomics, where changes are clearly event-driven, and lower thresholds for volatile areas such as cryptocurrency. Designing a more adaptive hybrid mechanism remains an open challenge for future work, but current thresholds effectively balance accuracy across domains.

## I    BROADER IMPACT

This paper introduces the *Social World Model* (SWM), a machine learning framework designed to analyze and forecast the evolution of population-level social beliefs using data from publicly available sources. On the positive side, such predictive models can support informed decision-making, anticipate societal shifts, and assist policymakers, journalists, and researchers in understanding public opinion trends. However, we also recognize the potential risks associated with misusing these technologies—for instance, influencing public perception through targeted messaging, reinforcing biases, or undermining democratic processes. While our dataset poses minimal privacy concerns, we urge responsible deployment and emphasize the need for ongoing ethical reflection. Our research is guided by a commitment to maximizing societal benefit while minimizing unintended consequences.

## J    CASE ANALYSIS

### J.1    CASE STUDY ON SUCCESSFUL CASES

For example, on January 22, 2024, the social belief regarding OpenSea's potential bankruptcy by March 1 suddenly surged from 2.5% to 50.0%. The social reasoner identified a possible cause: *Terraform Labs, the parent company of the now-defunct TerraUSD stablecoin, filed for Chapter 11 bankruptcy protection*. While this event is not directly linked to OpenSea, OpenSea had previously supported Terra-based NFTs, allowing users to trade them on its marketplace.

For example, when predicting whether OpenSea would announce a token by May 1, 2024, given historical data up to January 1, 2024, a time-series model consistently estimates the probability at 0.12. However, the actual probability collected from PolyMarket jumps to 0.5 in a single day. In contrast, our social world model identifies key social events, such as *BlackRock and VanEck updating their SEC filings for Bitcoin ETFs* and *Bitcoin surging above $46,000*, adjusting its prediction to 0.43. This demonstrates that the social world model can dynamically adapt to evolving social events, leading to more responsive and accurate forecasts.

## J.2 CASE STUDY ON FAILED CASES

We also identify failure cases of SWM in handling dramatic changes. For example, markets such as "Will Trump say 'Green New Scam' 3 or more times during a rally?" exhibit abrupt changes but do not represent meaningful shifts in collective belief, instead reflecting entertainment-driven betting. Similarly, in sports prediction markets, our model struggles due to its daily prediction granularity; sports outcomes often hinge on developments within hours. As a result, SWM fails on cases like "Will the Celtics win the 2024 NBA Finals 4-1?", where real-time feedback is critical. These examples illustrate limits in domains where belief changes are either trivial or too rapid. Sports are generally considered unsuitable for modeling with SWM due to their rapid change property.

## K   LIMITATIONS

Our approach has several limitations. First, the reasoning module of the agent is trained using annotations generated by GPT-4o rather than human-labeled reasoning traces. While GPT-4o produces high-quality outputs, these model-generated annotations may contain stylistic artifacts or patterns that differ from human logic. Second, our dataset is sourced from PolyMarket, an open prediction platform that, despite offering diverse and realistic data, may include noise or inconsistencies in market resolution, potentially introducing hidden biases or variance that affect model robustness and interpretability. Lastly, since both the dataset and reasoning traces rely on large language models, which can reflect societal biases present in their training data, there is a risk that the resulting agents may inherit or amplify these biases, raising concerns about fairness, representation, and unintended discriminatory behavior in real-world deployment.

