# OpenReview forum: "Building Social World Model with Large Language Models"
_ICLR.cc/2026/Conference — Submitted to ICLR 2026_

### Official Review · Reviewer_HWeU · 2025-10-30

**Soundness:** 2
**Presentation:** 3
**Contribution:** 2
**Rating:** 6
**Confidence:** 3

**Summary:**

This paper introduces the Social World Model (SWM), a novel LLM-based framework designed to model and predict the evolution of social beliefs over time in response to significant social events. The central idea is to treat social beliefs as states and social events as actions within a temporal world model, analogous to world models in reinforcement learning, but grounded in collective human opinion rather than physical environments. The model explicitly decouples social reasoning and belief transition, leveraging a posterior-guided reasoner (powered by GPT-4o) and an LLM-based transition model trained via ELBO optimization. To evaluate SWM, the authors construct SWM-Bench, a large-scale benchmark derived from over 300,000 samples of real-world Polymarket prediction market data across domains like politics, crypto, and sports. SWM significantly outperforms strong time-series and LLM baselines, particularly in cases of abrupt belief shifts.

**Strengths:**

1. The concept of modeling social beliefs as state transitions influenced by events is interesting and addresses an important societal challenge
2. SWM-Bench provides a valuable resource with 300,000+ real-world data points across diverse domains
3. The treatment of events as latent variables and use of ELBO optimization cleverly addresses the lack of paired state-action data

**Weaknesses:**

1. The core technical contribution is primarily applying existing ELBO optimization and LLM fine-tuning to a new domain rather than developing fundamentally new methods
2. The paper doesn't validate whether identified events actually cause belief changes or are merely correlated

**Questions:**

N/A

---

> ### Author Response · Authors · 2025-12-01
>
> # Novelty claim
> Our paper makes three key contributions:
>
> (1) **Formalizing the concept and modeling framework of Social World Models**.
> We are among the first to formalize the notion of a social world model—predicting shifts in collective beliefs conditioned on social events. Our formulation marginalizes over diverse social events and decomposes the problem into two components, social reasoning and social world modeling, as defined in Eq. (1) and Eq. (2).
>
> (2) **Introducing SWM-Bench, a large-scale benchmark for social belief prediction**.
> We construct SWM-Bench, a large-scale dataset derived from PolyMarket, to systematically evaluate models on social belief prediction tasks (Line 22–24). This benchmark provides the first standardized platform for studying social world models at scale.
>
> (3) **Developing a posterior-guided training and inference algorithm for SWM**.
> Building on our formulation, we propose a posterior-guided algorithm that integrates a social reasoner with an ELBO-based objective, enabling joint optimization of both components of the social world model (Line 326–336).
>
> Although ELBO losses and retriever-based reasoning are established techniques individually, our work introduces a fundamentally new application and integration of these tools for the novel task of social belief modeling. We define the core concepts, establish the benchmark, and provide the first systematic framework for this domain. Similar to prior works such as Hindsight [1], which also combine ELBO and retrieval components yet are considered novel, our contribution lies not in reusing techniques but in applying them in a new, impactful setting with new theoretical and empirical formulation.
>
> # Causality modeling
> Our method is **not intended to recover ground-truth causal relationships** between news and belief shifts. Because the data is purely observational—without interventions or counterfactual traces—establishing true causality is fundamentally impossible in this setting.
>
> In our formulation, the Social World Model learns a transition distribution $P(s' \mid s, a)$, where $a$ is a **predictive latent action** inferred from news content. This latent action serves as an explanatory abstraction of how external events influence belief states. Importantly, $a$ is **not guaranteed to correspond to a true causal mechanism**, and verifying its causal status is infeasible.
>
> However, this does **not** undermine our approach. As in standard world-modeling practice, such latent variables are designed to be *predictively useful*, not causally validated. The predictive latent action $a$ summarizes dominant drivers behind belief transitions and significantly improves forecasting accuracy—**which is the core objective of SWM**. Therefore, the absence of causal guarantees does not affect the applicability or effectiveness of our method.
>
>
> [1] Paranjape et al. Hindsight: Posterior-guided training of retrievers for improved open-ended generation.

---

### Official Review · Reviewer_zSoE · 2025-10-31

**Soundness:** 3
**Presentation:** 4
**Contribution:** 1
**Rating:** 2
**Confidence:** 4

**Summary:**

The paper introduces what the authors call a Social World Model — a framework in which Large Language Models (LLMs) are leveraged to model how collective social beliefs evolve in response to real-world events. The work focuses on predicting “social belief,” which is represented as a pair of yes/no questions along with the community expectation (i.e., the fraction of “yes” votes on the Polymarket prediction market).

The model approaches this prediction problem through an autoregressive formulation of the time series, introducing a latent event variable. Training is performed via a variational ELBO objective that involves several components the authors refer to as the social reasoner, posterior guide reasoner, and the social world model itself.

**Strengths:**

The paper presents a compelling narrative by introducing intuitive concepts to model large-scale social dynamics. The engineering effort required to obtain the reported results is impressive, as it involves training LLM-based components within a Bayesian approximate inference framework — a task that is technically challenging given the size of the models involved (e.g., the social reasoner and posterior networks). The authors skillfully incorporate both pretraining and fine-tuning within the ELBO framework to make this feasible.

**Weaknesses:**

Despite the engaging presentation and terminology, the core technical contribution of the paper is limited. From a machine learning perspective, the proposed approach is not conceptually novel. At its core, it is a time series latent variable model (reminiscent of earlier works such as  Chung et al. (2015), with the main difference being that LLMs are used as components of the recognition and generative networks. While this integration is interesting from an engineering standpoint, it does not represent a significant methodological advance for ICLR.

Furthermore, the claims made under the banner of a “Social World Model” seem overstated, given that the method ultimately performs time series prediction on Polymarket data. For these reasons, this work might be better suited for venues such as WWW or ICWSM, which focus on e.g., computational social science and large-scale data-driven analyses of online behavior.

*References:*

- Chung et al. *A Recurrent Latent Variable Model for Sequential Data*. NeurIPS (2015)

**Questions:**

1. Why do you refer "the reasoner" reasoner? In the context of LLMs, this term can be misleading, since “reasoning” is typically associated with chain-of-thought (CoT) or reinforcement learning on CoT approaches.

2. Why is one component fine-tuned while another uses LoRA adaptation? The choice seems inconsistent. Could you clarify the reasoning behind this design?

3. The manuscript refers to “*predicting how social beliefs evolve in response to events*”. For me the fundamental problem is that we don't know that the people in polymarket are reacting to *those news*, as opposed to other events. How do you address the risk of capturing spurious correlations instead of genuine causal responses?

---

> ### Author Response · Authors · 2025-12-01
>
> # Novelty claim
> Our paper makes three key contributions:
>
> (1) **Formalizing the concept and modeling framework of Social World Models**.
> We are among the first to formalize the notion of a social world model—predicting shifts in collective beliefs conditioned on social events. Our formulation marginalizes over diverse social events and decomposes the problem into two components, social reasoning and social world modeling, as defined in Eq. (1) and Eq. (2).
>
> (2) **Introducing SWM-Bench, a large-scale benchmark for social belief prediction**.
> We construct SWM-Bench, a large-scale dataset derived from PolyMarket, to systematically evaluate models on social belief prediction tasks (Line 22–24). This benchmark provides the first standardized platform for studying social world models at scale.
>
> (3) **Developing a posterior-guided training and inference algorithm for SWM**.
> Building on our formulation, we propose a posterior-guided algorithm that integrates a social reasoner with an ELBO-based objective, enabling joint optimization of both components of the social world model (Line 326–336).
>
> Although ELBO losses and retriever-based reasoning are established techniques individually, our work introduces a fundamentally new application and integration of these tools for the novel task of social belief modeling. We define the core concepts, establish the benchmark, and provide the first systematic framework for this domain. Similar to prior works such as Hindsight [1], which also combine ELBO and retrieval components yet are considered novel, our contribution lies not in reusing techniques but in applying them in a new, impactful setting with new theoretical and empirical formulation.
>
> # Difference with Chung et al.
> We emphasize that the connection to Chung et al. is limited to a shared use of the ELBO objective as a standard optimization tool for latent-variable models. Our main contribution lies in formulating social dynamics as a latent-variable problem and showing how to optimize it through an ELBO objective. ELBO is a widely used and well-established technique for latent-variable modeling, and adopting it does not diminish novelty. Indeed, related works such as [1] and [2] also build on ELBO-based formulations yet are still considered novel because their contributions lie in how latent variables are defined and where the method is applied—mirroring the role ELBO plays in our work.
>
> # Definition of reasoning
> The idea behind "reasoning" is that the model must attribute the primary driver of a belief change. We treat this as an attribution process, where the posterior distribution is used to estimate which latent action best explains the observed transition. In other words, after observing the posterior, the model must **reason** about the most plausible reason behind the change. This terminology is descriptive rather than essential, and does not affect the core technical contributions of our paper.
>
> # Causality modeling
> Our method is **not intended to recover ground-truth causal relationships** between news and belief shifts. Because the data is purely observational—without interventions or counterfactual traces—establishing true causality is fundamentally impossible in this setting.
>
> In our formulation, the Social World Model learns a transition distribution $P(s' \mid s, a)$, where $a$ is a **predictive latent action** inferred from news content. This latent action serves as an explanatory abstraction of how external events influence belief states. Importantly, $a$ is **not guaranteed to correspond to a true causal mechanism**, and verifying its causal status is infeasible.
>
> However, this does **not** undermine our approach. As in standard world-modeling practice, such latent variables are designed to be *predictively useful*, not causally validated. The predictive latent action $a$ summarizes dominant drivers behind belief transitions and significantly improves forecasting accuracy—**which is the core objective of SWM**. Therefore, the absence of causal guarantees does not affect the applicability or effectiveness of our method.
>
> # Training details
> As mentioned in Line 849-854, we fine-tune both social reasoner and social predictor under LoRA configuration. Therefore, both are aligned with each other and there is no misalignment.
>
>
> [1] Paranjape et al. Hindsight: Posterior-guided training of retrievers for improved open-ended generation.
>
> [2] Chen et al. Diffusion Forcing: Next-token Prediction Meets Full-Sequence Diffusion

---

### Official Review · Reviewer_tYsT · 2025-11-01

**Soundness:** 2
**Presentation:** 2
**Contribution:** 3
**Rating:** 4
**Confidence:** 4

**Summary:**

The paper introduces the Social World Model (SWM), a framework that models how social beliefs evolve in response to real-world events. Using data from Polymarket, the authors construct SWM-Bench, a large-scale benchmark for social belief prediction. SWM combines large language models with probabilistic event modeling to capture belief transitions over time and demonstrates improved performance over traditional time-series and LLM baselines.

**Strengths:**

Novel dataset creation
The introduction of SWM-Bench, derived from real-world Polymarket data, is a significant contribution. It provides a large-scale, domain-diverse benchmark for studying social belief dynamics and enables systematic evaluation of models that combine temporal forecasting with social reasoning. The dataset’s grounding in real market behavior gives it potential value for future research in social simulation and opinion modeling.

Innovative modeling framework
The proposed Social World Model (SWM) offers a creative formulation that treats social belief evolution as a state-transition process influenced by latent social events. The integration of LLM-based reasoning with probabilistic modeling (through ELBO optimization and posterior-guided training) is technically interesting. This framework bridges time-series prediction and social reasoning, contributing a fresh perspective to computational social science.

**Weaknesses:**

Limited justification for market-based data as authentic social opinion
The paper assumes that prediction market data, such as Polymarket transactions, are more reliable and authentic indicators of public opinion. However, this claim is not sufficiently supported. The argument would benefit from a clearer explanation of why financially motivated trading behavior better reflects social beliefs than alternative data sources like surveys or social media.

Potential bias and groupthink in Polymarket data
The use of Polymarket as the primary dataset introduces the risk of collective biases. Markets often exhibit herd behavior, social reinforcement, or correlated trading patterns. These dynamics may distort the very opinion signals that SWM aims to model. The paper does not discuss how such effects are mitigated or how they might affect model reliability.

Limited comparison to other reasoning-based forecasting methods
The paper primarily compares SWM against time-series models and simple LLM baselines, but not against retrieval-augmented or reasoning-based approaches. This omission makes it difficult to evaluate how SWM performs relative to more advanced forecasting frameworks.

Generalization limited to a single data source
The evaluation is confined to Polymarket data. Without evidence of cross-domain performance, it is unclear whether SWM generalizes to other platforms such as Good Judgment Project, Metaculus, or text-based datasets like OpinionsQA. This limits the claims of broad applicability.

**Questions:**

1. The authors claim that “free markets serve as a classical example of aggregated public opinions.” Could they elaborate on this reasoning? Specifically, what properties of market behavior make it a suitable proxy for collective belief formation?

2. How does SWM handle correlated or self-reinforcing behavior in markets, where traders may be influenced by prevailing trends or dominant narratives rather than independent judgment?

3. How would SWM compare to retrieval-augmented forecasting models, such as “Approaching Human-Level Forecasting with Language Models” (Halawi et al., 2024)?

4. Has the generalization of SWM been tested on other datasets or domains, such as expert forecasting platforms or broader opinion benchmarks (some examples of which are mentioned in weakness)?

---

> ### Author Response · Authors · 2025-12-01
>
> # Polymarket as data source
> We emphasize the unique advantage of Polymarket data for social dynamic modeling:
>
> 1. **Scale, liquidity, and market dominance**:
> Polymarket is the largest, most liquid real-world prediction market, with over $18B in trading volume in 2025, far surpassing alternatives such as Kalshi. High liquidity reduces noise, stabilizes prices, and ensures that market beliefs reflect contributions from a large, diverse participant base rather than a small or niche community.
>
> 2. **Broad coverage across domains**:
> As noted in Lines 137–159, Polymarket spans a wide range of topics—politics, macroeconomics, global events, crypto, sports, and more—providing substantially greater domain diversity than any single forecasting platform or survey dataset. This broad topical coverage makes it ideal for learning generalizable patterns of collective belief dynamics.
>
> 3. **Empirical cross-market consistency**:
> Prior work shows strong correlation between Polymarket and Kalshi on comparable markets [1], indicating that Polymarket prices are not idiosyncratic but align well with other independent forecasting communities. This cross-platform agreement strengthens the case that Polymarket reflects genuine aggregated beliefs rather than platform-specific artifacts.
>
> 4. **Incentive-aligned belief aggregation -- not like trading**:
> Polymarket is a prediction market—not a financial trading venue. Prices directly encode participants’ probability beliefs and are not shaped by institutional strategies, hedging, or large organizational actors. This makes the signals far closer to collective opinions than traditional market behavior.
>
> 5. **Correlation and herding are part of real belief dynamics**:
> Any correlated or trend-following behavior reflects how collective beliefs naturally evolve, not institutional trading pressure. Since SWM models predictive belief shifts, not causality, these patterns are meaningful signal rather than a flaw—they are precisely the social dynamics SWM is designed to capture.
>
> # Difference between prediction market and stock market
> Prediction markets embody a classical economic principle: **prices directly reflect the marginal belief of traders willing to stake money on a specific event outcome**. This principle does *not* hold in modern stock markets, where prices are shaped by quantitative algorithms, hedging flows, liquidity constraints, macroeconomic shocks, and corporate fundamentals—factors largely unrelated to belief about any single event. In contrast, PolyMarket contracts have payoffs tied *only* to the outcome of one well-defined question, creating a clean, incentive-aligned link between belief and price.
>
> This mechanism encourages honest and independent belief expression: incorrect beliefs are immediately penalized, while accurate information is rewarded. Prediction markets have been repeatedly validated in economics and forecasting research as one of the most effective tools for aggregating dispersed information. Unlike surveys or social media, which are susceptible to noise, strategic reporting, or low-confidence opinions, PolyMarket’s design naturally filters out unreliable signals and captures a high-quality representation of collective belief.
>
> # Collective bias or herd behavior in Polymarket
> Collective bias is an inherent and expected property of all real-world prediction markets. Liquid markets like PolyMarket are still highly informative because market-making mechanisms (e.g., LMSR) and large trading volume naturally dampen harmful herd effects. More importantly, SWM is designed to model these collective dynamics rather than eliminate them: correlated or trend-driven shifts are part of how real beliefs evolve. Since SWM focuses on predicting belief transitions—not recovering ground-truth causality—such collective behavior is not a flaw but a fundamental aspect of the phenomenon we aim to capture.

---

> > ### Author Response · Authors · 2025-12-01
> >
> > # Additional baselines
> > We compare our method against RAG reasoning-based baselines, including approaches similar to [2]. Because the fine-tuned GPT model used in [2] is not publicly available, we implement similar baselines ourselves using the same base model using RL (GRPO) settings. The baselines allow the model to perform reasoning and text-based prediction in a manner consistent with [2]. Across all markets, their performance remains consistently worse than SWM.
> >
> > For the comparison with expert level forcast models, we cannot reach such high quality forcasting due to limited model size, but we emphasize that we aimed for a general framework to serve as an approach for modeling social dynamics.
> >
> > | Settings | Politics RMSE | Politics MAE | Sports RMSE | Sports MAE | Crypto RMSE | Crypto MAE | Election RMSE | Election MAE | Other RMSE | Other MAE |
> > | -------- | ------------- | ------------ | ----------- | ---------- | ----------- | ---------- | ------------- | ------------ | ---------- | --------- |
> > | GRPO     | 0.4131        | 0.3850       | 0.3771      | 0.3586     | 0.4035      | 0.3603     | 0.4468        | 0.3965       | 0.3695     | 0.3363    |
> > | SWM      | 0.3340        | 0.2935       | 0.2696      | 0.2361     | 0.2530      | 0.2115     | 0.3285        | 0.2885       | 0.2491     | 0.1960    |
> >
> >
> > # Generalization of SWM
> > We conduct cross-domain train-test to evaluate the generalization ability of our model. We observe that the social world model demonstrates surprisingly strong generalization across domains. This suggests that it learns underlying social patterns that govern the relationships between social states and events. These findings highlight the potential for developing a foundational social world model capable of understanding and reasoning across multiple domains, paving the way for more comprehensive and adaptable social prediction systems.
> >
> > For evaluation on markets besides Polymarket, we refer to the section of **[Polymarket as data source]** to show that similar topics in both Polymarket and Kalshi have highly correlated results due to high volume of them.
> >
> > | **Train \\ Test** | **crypto** | **politics** | **election** | **sports** | **other** |
> > |-------------------|------------|--------------|--------------|------------|-----------|
> > | **crypto**        | 0.28       | 0.33         | 0.34         | 0.27       | 0.27      |
> > | **politics**      | 0.28       | 0.33         | 0.30         | 0.29       | 0.27      |
> > | **election**      | 0.26       | 0.31         | 0.32         | 0.29       | 0.27      |
> > | **sports**        | 0.29       | 0.33         | 0.34         | 0.31       | 0.26      |
> > | **other**         | 0.30       | 0.35         | 0.34         | 0.29       | 0.28      |
> >
> >
> >
> > [1] Ng et al. Price Discovery and Trading in Prediction Markets
> >
> > [2] Halawi et al. Approaching Human-Level Forecasting with Language Models

---

### Official Review · Reviewer_g4X5 · 2025-11-01

**Soundness:** 2
**Presentation:** 3
**Contribution:** 2
**Rating:** 4
**Confidence:** 4

**Summary:**

This paper proposes the Social World Model (SWM), a framework for predicting how social beliefs evolve in response to events using Large Language Models (LLMs). SWM treats events as latent variables and performs optimization using Evidence Lower Bound (ELBO) with posterior guidance. To evaluate SWM, the authors introduce SWM-Bench, a benchmark derived from Polymarket data containing over 300,000 samples across cryptocurrency, elections, sports, and politics.

**Strengths:**

S1. This paper tackles an important problem of modeling social belief dynamics, which has practical applications in event forecasting, policy making, and understanding community behavior. The weakly-supervised approach using only time-series data without explicit event-belief annotations makes the problem more tractable.

S2. SWM-Bench provides a valuable benchmark for social belief prediction given its substantial scale and diversity across multiple domains. This dataset could facilitate future research in this area.

S3. This paper conducts comprehensive ablation studies examining model size, window size, and event space size. Table 2 and Figures 3, 4, and 5 provide useful insights into component contributions and design choices.

**Weaknesses:**

W1. The technical novelty is limited, as SWM primarily combines existing techniques including Transformer encoders for time-series, standard ELBO optimization, and contrastive learning objectives. The main contribution appears to be adapting these methods to social belief prediction rather than methodological innovation.

W2. The evaluation is restricted to a single data source from Polymarket, which might have inherent biases in user demographics and market selection. Generalization to other social belief data sources remains unclear.

W3. This paper uses GPT-4o as the posterior-guided reasoner, making the approach dependent on proprietary models and potentially expensive for scaling. The computational costs and practical deployment considerations are not thoroughly discussed.

W4. Several technical details require clarification. The conditional independence assumption in Equation 1 is strong but justified only briefly. Based on Equations 6 and 7, the retriever implementation lacks details about retrieval methods and how similarity is computed.

**Questions:**

Q1: How does SWM perform on cross-domain generalization, such as training on political data but testing on crypto markets? This would better demonstrate SWM’s ability to capture general social dynamics versus domain-specific patterns.

Q2: The conditional independence assumption in Equation 1 seems critical but potentially problematic. Will there be some cases where beliefs are interdependent?

Q3: For Equations 6 and 7, what similarity metrics are used for retrieval?

Q4: How does SWM handle cases where important events are missing from the collected new data? What percentage of belief shifts in the test set can be attributed to events in news corpus?

---

> ### Author Response · Authors · 2025-12-01
>
> # Novelty claim
> Our paper makes three key contributions:
>
> (1) **Formalizing the concept and modeling framework of Social World Models**.
> We are among the first to formalize the notion of a social world model—predicting shifts in collective beliefs conditioned on social events. Our formulation marginalizes over diverse social events and decomposes the problem into two components, social reasoning and social world modeling, as defined in Eq. (1) and Eq. (2).
>
> (2) **Introducing SWM-Bench, a large-scale benchmark for social belief prediction**.
> We construct SWM-Bench, a large-scale dataset derived from PolyMarket, to systematically evaluate models on social belief prediction tasks (Line 22–24). This benchmark provides the first standardized platform for studying social world models at scale.
>
> (3) **Developing a posterior-guided training and inference algorithm for SWM**.
> Building on our formulation, we propose a posterior-guided algorithm that integrates a social reasoner with an ELBO-based objective, enabling joint optimization of both components of the social world model (Line 326–336).
>
> Although ELBO losses and retriever-based reasoning are established techniques individually, our work introduces a fundamentally new application and integration of these tools for the novel task of social belief modeling. We define the core concepts, establish the benchmark, and provide the first systematic framework for this domain. Similar to prior works such as Hindsight [1], which also combine ELBO and retrieval components yet are considered novel, our contribution lies not in reusing techniques but in applying them in a new, impactful setting with new theoretical and empirical formulation.
>
>
> # Polymarket as data source
> We emphasize the unique advantage of Polymarket data for social dynamic modeling:
>
> 1. **Scale, liquidity, and market dominance**:
> Polymarket is the largest, most liquid real-world prediction market, with over $18B in trading volume in 2025, far surpassing alternatives such as Kalshi. High liquidity reduces noise, stabilizes prices, and ensures that market beliefs reflect contributions from a large, diverse participant base rather than a small or niche community.
>
> 2. **Broad coverage across domains**:
> As noted in Lines 137–159, Polymarket spans a wide range of topics—politics, macroeconomics, global events, crypto, sports, and more—providing substantially greater domain diversity than any single forecasting platform or survey dataset. This broad topical coverage makes it ideal for learning generalizable patterns of collective belief dynamics.
>
> 3. **Empirical cross-market consistency**:
> Prior work shows strong correlation between Polymarket and Kalshi on comparable markets [1], indicating that Polymarket prices are not idiosyncratic but align well with other independent forecasting communities. This cross-platform agreement strengthens the case that Polymarket reflects genuine aggregated beliefs rather than platform-specific artifacts.
>
> 4. **Incentive-aligned belief aggregation**:
> Polymarket is a prediction market—not a financial trading venue. Prices directly encode participants’ probability beliefs and are not shaped by institutional strategies, hedging, or large organizational actors. This makes the signals far closer to collective opinions than traditional market behavior.
>
> 5. **Correlation and herding are part of real belief dynamics**:
> Any correlated or trend-following behavior reflects how collective beliefs naturally evolve, not institutional trading pressure. Since SWM models predictive belief shifts, not causality, these patterns are meaningful signal rather than a flaw—they are precisely the social dynamics SWM is designed to capture.
>
>
> # Cross-domain generalization
> We conduct cross-domain train-test to evaluate the generalization ability of our model. We observe that the social world model demonstrates surprisingly strong generalization across domains. This suggests that it learns underlying social patterns that govern the relationships between social states and events. These findings highlight the potential for developing a foundational social world model capable of understanding and reasoning across multiple domains, paving the way for more comprehensive and adaptable social prediction systems.
>
> | **Train \\ Test** | **crypto** | **politics** | **election** | **sports** | **other** |
> |-------------------|------------|--------------|--------------|------------|-----------|
> | **crypto**        | 0.28       | 0.33         | 0.34         | 0.27       | 0.27      |
> | **politics**      | 0.28       | 0.33         | 0.30         | 0.29       | 0.27      |
> | **election**      | 0.26       | 0.31         | 0.32         | 0.29       | 0.27      |
> | **sports**        | 0.29       | 0.33         | 0.34         | 0.31       | 0.26      |
> | **other**         | 0.30       | 0.35         | 0.34         | 0.29       | 0.28      |

---

> ### Author Response · Authors · 2025-12-01
>
> # Posterior-guided reasoner cost
> The posterior-guided reasoner provides the attributed explanations behind belief changes—information that is extremely difficult and costly to collect directly. GPT-4o offers a strong and inexpensive approximation of this distribution: generating the full set of offline posterior explanations cost under \$100. We obtain this distribution using a single, carefully designed prompt that elicits the model’s estimated posterior over possible reasons.
>
> # Conditional independence assumption
> To better train and inference implementation, we adopt a conditional independence assumption. At each time step, the belief space $S_t$ may involve more than 1,000 time series, which far exceeds the context window of current LLMs. Modeling the full joint distribution $P(S_t \mid S_{<t})$ is therefore infeasible. For efficiency, we approximate independence and restrict the conditioning set to the most relevant prior signals: $top_k(S_{<t}) \subseteq S_{<t}$. This reduces the effective input size, enables parallel training, and keeps optimization within feasible LLM context limits. In short, the conditional independence assumption is a necessary simplification to make the social world model computationally manageable.
>
> # Retriever details
> The retriever is conducted with a basic `voyage` model to embed the questions for different markets. We just use embedding-based similarity for retrieve related markets based on questions.
>
>
> # Missing event cases
> It is true that some cases provide little usable signal. When none of the candidate events receive a non-zero posterior probability from the posterior-guided reasoner, we simply skip those instances. Such data contains no informative attribution signal and therefore does not contribute meaningfully to model learning.
>
> [1] Paranjape et al. Hindsight: Posterior-guided training of retrievers for improved open-ended generation.
>
> [2] Ng et al. Price Discovery and Trading in Prediction Markets

---

### Author Response · Authors · 2025-12-01

# General Response
We appreciate the reviewers’ efforts in evaluating our paper. Below, we summarize the key points reviewers raised—items marked with ** indicate issues for which we provide additional experiments or clarifications, while unmarked items reflect strengths acknowledged by them. "Action/Summary" includes the highly summarized rebuttal content for each reviewer.
|                                  | Reviewer g4X5                                                                    | Reviewer tYsT                                               | Reviewer zSoE                                                                    | Reviewer HWeU                                                                                                     | Action/Summary                                                                                                                                                                                                                                 |
| -------------------------------- | -------------------------------------------------------------------------------- | ----------------------------------------------------------- | -------------------------------------------------------------------------------- | ----------------------------------------------------------------------------------------------------------------- | ---------------------------------------------------------------------------------------------------------------------------------------------------------------------------------------------------------------------------------------------- |
| Novelty                          | "tackles an important problem", \*\***"The technical novelty is limited,..."**      | "Innovative modeling framework..."                          | \*\***"core technical contribution of the paper is limited..."**                     | \*\***"The core technical contribution is primarily applying..."**                                                    | `summary`: 2 reviewers mentioned that our idea is novel. `Rebuttal for Reviewer g4X5, zSoE, HWeU`: We emphasize our contribution is (1) the concept and modeling methods of social world model; (2) a benchmark for SWM and (3) training/inference algorithm for SWM. ELBO serves as a core modeling method. |
| Resource                         | "provides a valuable benchmark..."                                               | "Novel dataset creation..."                                 | NA                                                                               | "SWM-Bench provides a valuable resource..."                                                                       | `summary`: 3 reviewers agree that our benchmark is valuable.                                                                                                                                                                                   |
| Generalization | \*\***"Generalization to other social belief data sources remains unclear..."**      | \*\***"Generalization limited to a single data source..."**     | NA                                                                               | NA                                                                                                                | `Rebuttal for Reviewer g4X5, tYsT`: We emphasize the broadness of Polymarket data and conduct cross-domain evaluation to prove the generalization ability of SWM.                                       |
| Causality                        | NA                                                                               | NA                                                          | \*\***"...capturing spurious correlations instead of genuine causal responses"**      | \*\***"...doesn't validate whether identified events actually cause belief changes or are merely correlated..."**      | `Rebuttal for Reviewer zSoE & HWeU`: We emphasize that the *a* in $P(s' \mid s, a)$ serves as a predictive latent action. Our news data is purely observational, so guarantees of causality are impossible; predictive abstractions suffice for SWM. |

We also provide detailed clarifications on related concepts in response to the reviewers’ feedback, including training procedures, retriever design, missing-event handling, free market assumption, and the computational cost of the reasoners.

---

### Meta-Review · Area_Chair_xR7f · 2026-01-08

**Summary:**

The paper’s main contribution is the introduction of a novel dataset. Overall, the referees recognized the relevance of this contribution, though there was a general consensus that the paper, in its current form, may not yet be ready for acceptance.
The primary concern relates to the limited technical depth of the current modeling approach. While the dataset itself is an asset for advancing research on Social World Models, much of the accompanying methodology appears relatively straightforward and closely aligned with standard time-series forecasting techniques.
The paper would benefit from further development of the underlying models to more clearly articulate and address the distinctive aspects of Social World Models. With additional technical elaboration  the work has the potential to make impactful contributions to the AI community

**Reviewer Concerns:**

Apart from contribution being modest, all other comments seem to be partially addressed.

**Reviewer Scores:**

Not sure.

---

### Decision · Program_Chairs · 2026-01-26

Reject